# 1 The weather diaries of the Kirch family: Leipzig, Guben, and Berlin, 1677-
# 2 1774

Stefan Brönnimann*, Yuri Brugnara
*Oeschger Centre for Climate Change Research and institute of Geography, University of Bern*
*corresponding author: Stefan Brönnimann, stefan.broennimann@giub.unibe.ch
**Abstract**
Astronomer and calendar maker Gottfried Kirch was a keen weather observer and made weather notes in his
diary starting in 1677 in Leipzig. In parallel, his second wife Maria Margaretha Winkelmann started a weather
diary in 1700 in Berlin. The diaries also contain instrumental measurements of temperature and later pressure.
After the death of Gottfried in 1710 and Maria Margaretha in 1720, observations were continued by their son
Christfried and then for another 44 years by their daughter Christine. The last measurements date to 1774.
Together, the diaries span almost a century of weather observations. The instrumental measurements constitute
the oldest part of Germany's longest temperature series, which was however only available as monthly means up
to now. Here we publish the imaged diaries, together amounting to 10445 images. Further, we present the
digitised instrumental series, which will serve as the starting point for a new, daily Berlin series. By comparing
the series to neighbouring records, we show that the pressure data are reliable in a quantitative sense, whereas
this is true for the temperature data only in a qualitative sense as the temperature scale was not converted.

## 20 1. Introduction

Long historical climate records are invaluable to better understand variations in climate and the
underlying mechanisms. For a long time, the emphasis was on monthly or seasonal averages, and long
meteorological series were often only available in that form. Recently, changes in weather patterns and
extremes came into focus. New tools such as reanalyses (e.g., Slivinski et al., 2019) or analog
approaches (Pappert et al., 2022) now allow the daily weather to be reconstructed, from which
important conclusions can be drawn about decadal to multidecadal variations in weather as well as
extreme weather (Brönnimann, 2022). However, many of the long series are not available at daily or
sub-daily resolution (i.e., the individual measurements), but only as monthly means. It is therefore
often required to revisit archives, image and digitise the sub-daily data and start the homogenisation
processes anew.
In this paper, we present the work of imaging, digitising, and processing for the case of the longest
German record, that from Berlin which goes back to 1701. Specifically we analyse the weather diary
of the Kirch family, covering 1677-1774, which contains some instrumental observations in 1697 and
then from 1701 onward. This record has been widely used since the late 18[th] century. Karl-Ludwig
Gronau compiled the measurements and calculated monthly means (Gronau, 1807), supplementing the
sometimes sparse measurements in an unknown way. In the 19[th] century, Johann Heinrich Mädler
continued working on a Berlin series and presented a new, extended series (Mädler, 1825). Hellmann
(1893) re-discovered the weather diaries of Maria Margaretha Kirch, Lenke (1964) used the Berlin
data in his study on the cold winter 1708/9 and Pelz (1978) examined the Kirch diaries. However,
most other authors did not consult the original diaries. In the German Democratic Republic, Bahr
(1966) worked on the history of the Berlin series in the context of her dissertation. Subsequent work
led to the publication of a daily temperature series back to 1766, which was recently digitized by
Kadow et al. (2016). In Western Germany, Pelz (1997) re-homogenised and published the monthly
Berlin series (Cubasch and Kadow, 2010). What is still missing is the daily or sub-daily temperature
series earlier than 1766. Furthermore, although pressure was also measured, it was never digitised or
analysed. Therefore, we revisited the original sources, imaged the diary and digitised most of the
instrumental measurements (both temperature and pressure) contained in the sheets.
The paper is organised as follows. Section 2 provides background about the Kirch family and their
meteorological observations. Sect. 3 describes the diary and its history. In Sect. 4 we describe the
digitising, Quality Control, and comparison with other sources. Results are presented in Section 5 and
conclusions are drawn in Sect. 6.
**2. The Kirch family**
*2.1. Life and work*
Gottfried Kirch was one of the leading astronomers of the late 17[th] century. A recent biography
(Herbst, 2022) gives a detailed account of his life and work, which is only briefly summarized in the
following. Kirch was born in Guben (Fig. 1). In the 1660s he started to publish calendars, which
remained an important source of income for the family even after his death. As an astronomer, Kirch
became famous in the early 1680s when he discovered a comet and the star cluster M11. His second
wife Maria Margaretha Winkelmann also was an astronomer (but she was not admitted to the
university of Halle, to which she applied). She discovered the comet C/1702 H1 and worked on
sunspots. Gottfried and Maria Margaretha Kirch had six children, many of which supported or
continued the astronomical and meteorological work. Two of them, Christfried and Christine, had their
own weather diaries. Christine was further supported by her sisters Margaretha and Dorothea. Figure 2
show the observers and observation locations on a time axis.
The Kirch family lived in Leipzig from 1676 onward, then moved to Guben in 1692 (Fig. 1). In 1700
Gottfried Kirch was appointed as Royal astronomer by the newly funded Prussian Academy of
Science in Berlin (with Gottfried Wilhelm Leibniz as president). However, the astronomical
observatory was not built yet, and for the next 10 years observations were performed in the family's
apartment (Fig. 1).
Also the promised apartment was not ready in 1700, and therefore the family initially stayed at
different places. In 1708 they eventually moved into the „Astronomenhaus" at Dorotheenstrasse 10
(Fig. 1). The observatory was just next door (Fig. 3), but was officially opened only in 1711, although
observations were made earlier.
Gottfried Kirch died in 1710. After the death of her husband, Maria Margaretha Kirch and her son
Christfried Kirch continued the observations. They decided that Christfried should note the
astronomical observations and Maria Margaretha Kirch the weather observations. However, the
financial situation became more and more precarious for Maria Margaretha Kirch. Despite her
qualification, she could not follow as a director of the observatory but continued to publish the
calendars. In 1712, she moved into the house of Baron von Krosigk at Wallstrasse 135 (Fig. 1), where
she observed for the next two years (however, there was no thermometer). In 1714, she moved to
Gdansk with the children. In 1716, Christfried Kirch became the director of the Berlin observatory,
and Maria Margaretha Kirch moved back to her son into the „Astronomenhaus". In 1716 she began to
measure again and continued almost until her death on 29 December 1720. The data presented in this
article start in 1721 and were presumably all taken at the "Astronomenhaus" (location B in Fig. 1).
Measurements were continued together by Christfried und Christine Kirch. Christfried died in 1740.
Although not officially a member of the Prussian Academy, Christine continued the astronomical
observations and was paid by the Academy. She also performed meteorological measurements and
continued publishing the calendars. Her house was a gathering point for scientists of the 18th century.
Leonhard Euler was a frequent visitor, other guests include Joseph-Nicolas Delisle and Anders
Celsius. The last instrumental meteorological measurements date to 30 April 1774. Christine Kirch
died in 1782.

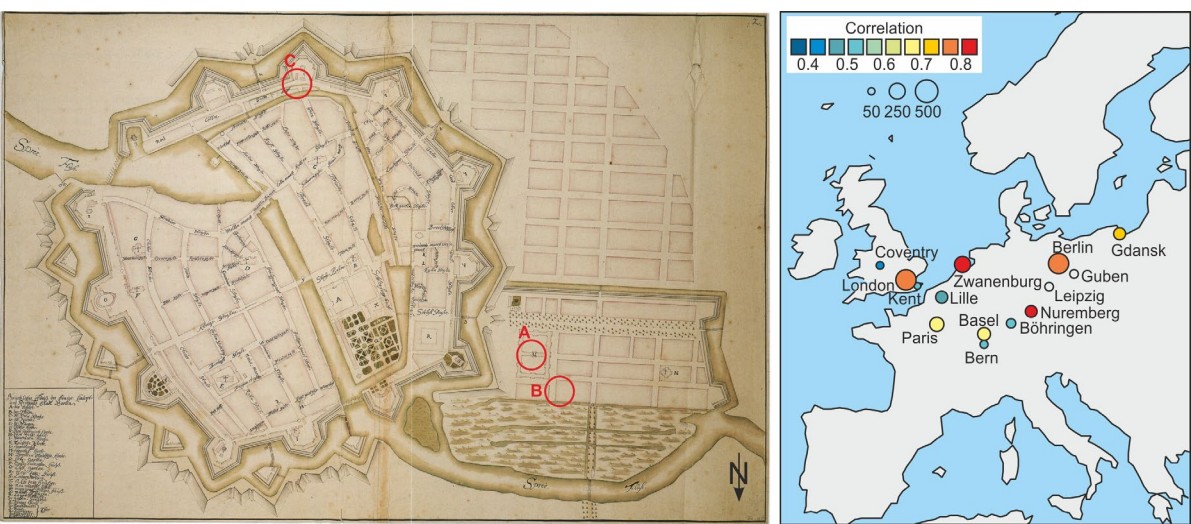


**Fig. 1.** Left: Map of Berlin in 1712. A: Astronomical observatory. B: Apartment of the Kirch family until 1712
and from 1716 onward ("Astronomenhaus"). C: House of Baron von Krosigk, where Maria Margaretha observed
from 1712-1714 (from map "Grundlicher Abriß, der königl. Haupt- und Residenz Stadt Berlin", unknown
author, public domain, wikimedia commons). Right: Locations of other weather observations in the Kirch diaries
(Berlin and empty circles) and other locations, coloured according to the Pearson correlation of their pressure
records with the Kirch measurements, Berlin, 1730-1770, in the overlapping period (circle area indicates tne
number of measurements *n*). For Berlin, the circle shows the correlation with the reconstruction EKF400v2.

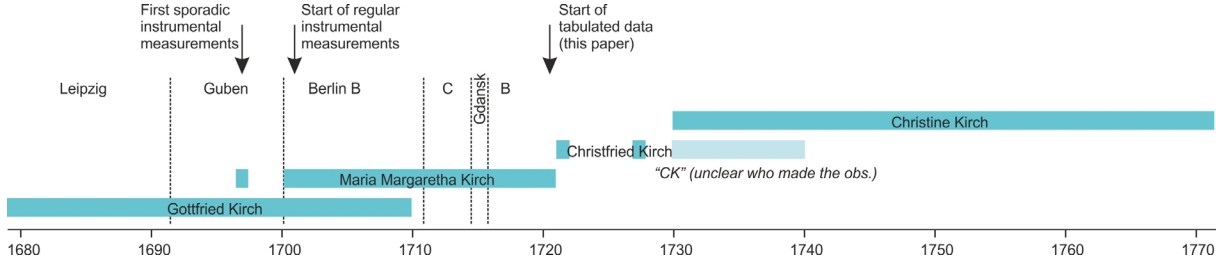

**Fig. 2.** Temporal overview of observers and observation locations.

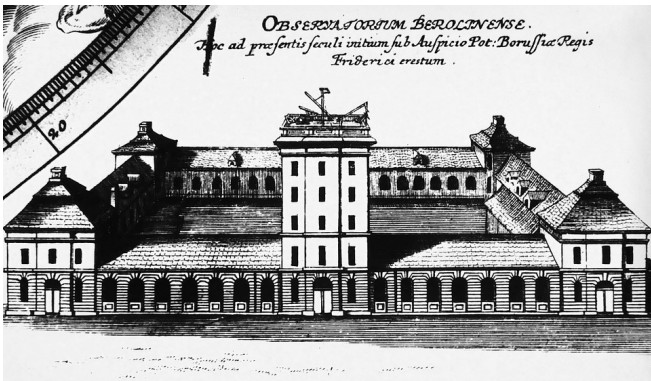

**Fig. 3.** The old Berlin astronomical observatory (excerpt from an etching published in the Atlas Coelestis of Johann Gabriel Doppelmayr, 1742, public domain, wikipedia).

*2.2. Instruments*

Gottfried and Maria Margaretha Kirch had a thermometer ("Wetterglas") since 1691. It was a Florentin-type thermometer manufactured by Gottfried Kirch: a glass bulb filled with spirit of wine with a closed tube. However, the thermometer seems to have been broken at some point. The notes clearly speak of an old and a new thermometer, although it is not fully clear when the change took place.

At that time observers made their own temperature scales as there was no agreed-upon scale. According to the description by Kirch, the thermometer had a 60 degree scale. The freezing point was at 20 degrees, and 40 degrees corresponded to a hot summer day. The temperature scale was later analysed by Lenke (1964), who converted the Berlin temperature data for the winter 1708/9 to a Celsius scale, though not without difficulties. Later, Christfried und Christine Kirch reportedly used a Fahrenheit thermometer, but the given temperatures are inconsistent with a Fahrenheit scale. Pelz (1978) mentions six different scales that were used in the Kirch diaries, of which only the first one is approximately known.

A barometer was in use since 1709. However, no details are known about the instrument. Likewise, not much is known about the siting of the instruments. According to Lenke (1964), measurements in the "Astronomenhaus" were made in a north facing window in the middle floor.

## 3. The weather diary

An overview of the weather diaries available for this study is given in Table 1. Gottfried Kirch's diary starts in 1677 when he worked in Leipzig. It contains mostly astronomical observations, but also sporadic weather information that were important for his astronomical observations. Noteworthy is Maria Margaretha's diary, a specific weather diary which starts in August 1700 (plus January to June 1697, containing instrumental measurements made in Guben). The first instrumental measurements in Berlin date to 18 January 1701 (Fig. 4), the day of the coronation of the Prussian king Friedrich I. For several years, the diaries of Gottfried and Maria Margaretha run parallel. The motivation behind the instrumental measurements most likely was checking the calendar (Herbst, 2022).

**Table 1.** Boxes with printouts of the Kirch diaries at Free University of Berlin, content of the boxes, original archive, number and availability of microfilms. Note that this Table also corresponds with the folder structure on the repository. More information on individual years is given in Pelz (1978).

| Author | Content | Original archive | Film number | Microfilms |
|---|---|---|---|---|
| Gottfried Kirch | astronomical diary 1677-1685 | Paris Observatory | Film 649I | not found |
| Gottfried Kirch | astronomical diary 1685-1689 | Paris Observatory | Film 649II | available |
| Gottfried Kirch | astronomical diary 1696-1704 | Paris Observatory | Film 650I | available |
| Gottfried Kirch | astronomical diary 1704-1708 | Paris Observatory | Film 650II | available |
| Gottfried Kirch | astronomical diary 1708-1710 | Paris Observatory | Film 650III | available |
| Maria Margaretha Kirch | weather diary 1697, 1700-1718 | Royal Observatory | Rolle 1/Film 583 Rolle 2/Film 584 | not found |
| Maria Margaretha Kirch and Christfried Kirch | weather diary 1718-1720, 1721 and 1728 | Royal Observatory | Rolle 3/Film 585 | not found |
| Christine Kirch | weather diary 1730-1734 | Royal Observatory | Rolle4/Film586 | available |
| Christine Kirch | weather diary 1734-1737 | Royal Observatory | Rolle5/Film587 | available |
| Christine Kirch | weather diary 1738-1743 | Royal Observatory | Rolle6/Film588 | available |
| Christine Kirch | weather diary 1743-1747 | Royal Observatory | Rolle7/Film589 | available |
| Christine Kirch | weather diary 1748-1756 | Royal Observatory | Rolle8/Film590 | available |
| Christine Kirch | weather diary 1757-1761 | Royal Observatory | Rolle9 | available |
| Christine Kirch | weather diary 1762-1770 and 1774 | Royal Observatory | Rolle 10/Film 592 Rolle 11/Film 593 | available |

Both the diaries of Gottfried and Maria Margaretha end only shortly before their deaths. After 1720, weather data were contained in Christfried's diary, but we only have data for selected years. From 1730 on the weather observations are noted in Christine's diary. Note that an attribution is difficult to make since both signed their observations with "CK". Most of the observations in our digitised record are from Christine, who was assisted by her sisters. One of the last pages of her diary, from 1770, is displayed in Fig. 5 (there are no entries at all for 1771-1773, and in 1774 only from January to April).

The diary of Gottfried Kirch was already famous in the 18[th] century. Several copies must exist. According to Lenke (1964), Joseph-Nicolas Delisle bought Gottfried Kirch's diary from Christine

Kirch and gave it to the "Dépôt général de la Marine". After the French Revolution, the diaries ended
up at the Observatoire de Paris where they remain to the present day. The diaries of Maria Margaretha,
Christfried and Christine Kirch are stored today at the Royal Observatory in Edinburgh. How they
ended up there is not known. It seems that for a long time, the location of these diaries was unknown.
Hellmann (1893) reports how he searched for them and how they eventually were found in Edinburgh.
He then published a transcription of the first years of the diary of Maria Margaretha (Hellmann, 1893).

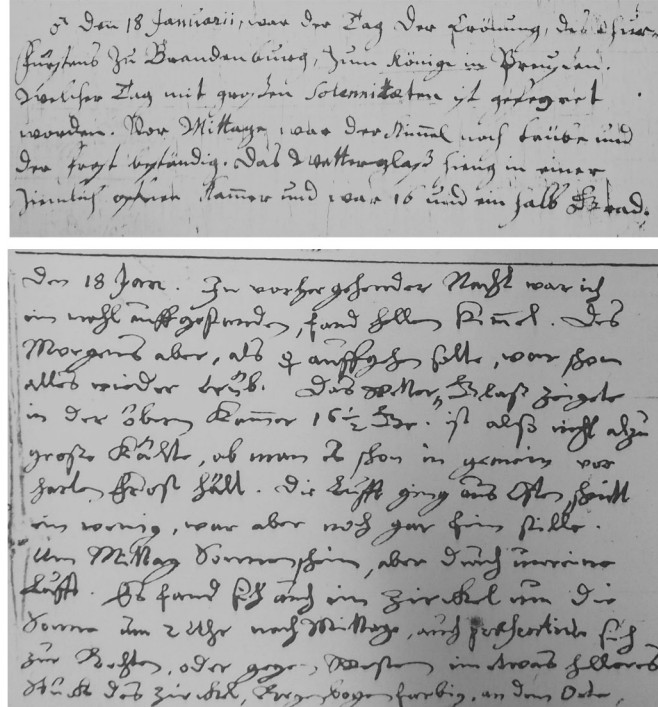


**Fig. 4.** First temperature measurement from Berlin on 18 January 1701, from the weather diary of Maria
Margaretha Kirch (The Crawford Collection of the Royal Observatory Edinburgh, top) and Gottfried Kirch
(Paris Observatory, bottom). Both mention that the thermometer was at 16.5 degrees. The last lines of the top
excerpt read: "Das Wetterglas hing in einer ziemlich offenen Kammer und war 16 und einhalb Grad." (The
weather glass was in an open chamber and was 16 and one half degrees).
At the request of the German Weather Service, the Royal Observatory in Edinburgh filmed the Kirch
diaries in 1959, and in 1962 the Paris Observatory filmed diaries of Gottfried Kirch. In 1977, 16 roles
of film were duplicated and Xerox print-outs were made. At this occasion, Pelz (1978) analysed the
diaries. The print-outs and films are still stored today at the library of the Institute of Meteorology of
the Free University of Berlin (Fig. 6). Of the 16 roles of film, only 12 could be found (missing is film
649I of Gottfried Kirch's diary and roles No. 1, 2, 3, of Maria Margaretha Kirch's diary, see Table 1).
Print-outs of all films are available.
The print-outs were imaged on 6-12 Mar 2020 by the first author. The paper quality did not allow an
automatic scanning. Hence the paper sheets were photographed with a handheld smartphone. The
impending lockdown due to the COVID pandemic forced us to work quickly. It was decided not to

image the part from 1677-1700 since it did not contain instrumental observations, and the ca. 7500 pages from 1700 to 1774 were photographed rather quickly. The imaged data sheets could then be transcribed during the lockdown and following period. In summer 2022, the first author returned to Berlin to also photograph the remaining portion (1677-1700). No data were transcribed from these images, they were merely made to have the diaries imaged in their entirety and to keep the diaries together electronically at one location.

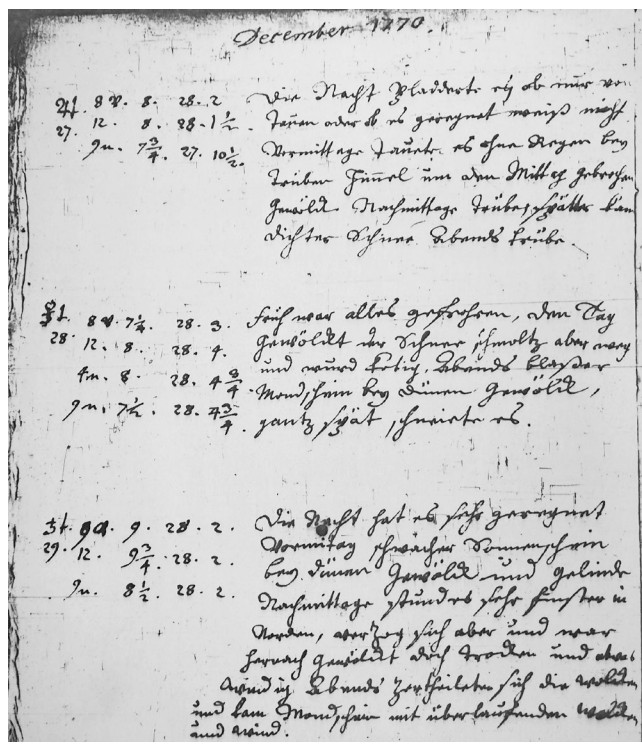

**Fig. 5.** Temperature and pressure measurements in 1770 by Christine Kirch (The Crawford Collection of the Royal Observatory Edinburgh).

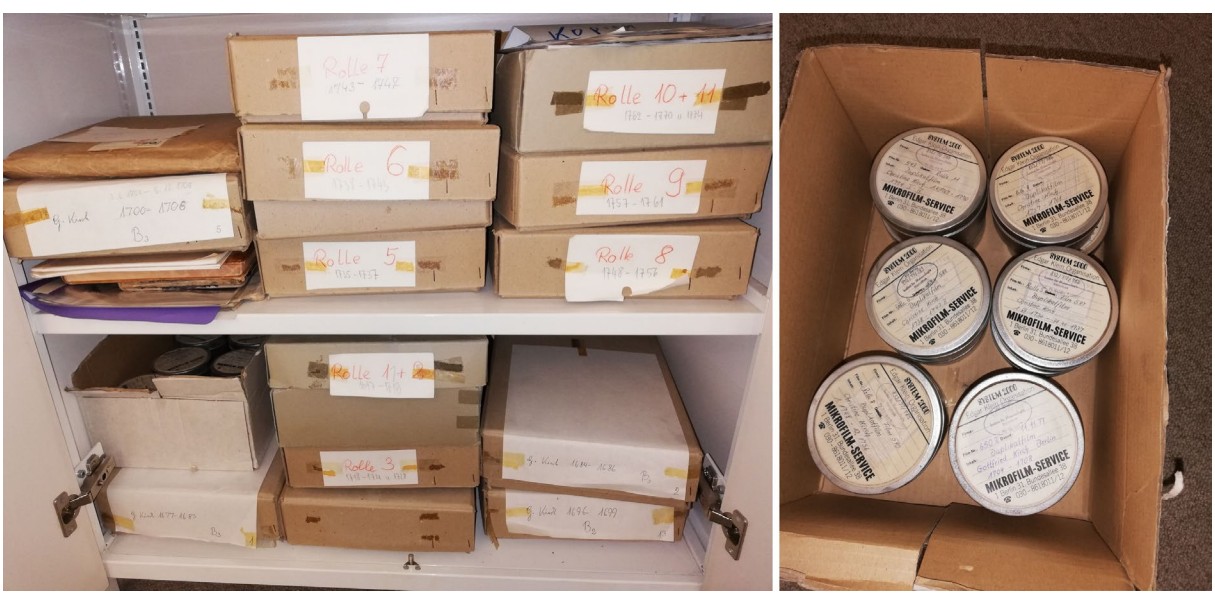

**Fig. 6.** Left: The boxes with Xerox print-outs photographed at Free University Berlin. Right: 12 Boxes with microfilms from which the print-outs were made.

## 4. Methods

Although first observations already start in the late 17th century, they were often not presented in a tabular form. Measurements were at times quite frequent, but they were often interspersed in the text, such as in Fig. 4. Sometimes measurements were organised in tables, but not consistently. We digitised the data only from 1720 onward, when they are presented in a consistently tabular format. The published images accompanying this article allow digitising also the earlier measurements.

We digitised temperature and pressure measurements. The actual keying work was performed by students of the University of Bern, who received a template and instructions (see spreadsheets in Supplementary Data: Tab "Info" contains the instructions).

The pressure data were then processed as described in Brugnara et al. (2020) and converted to the station exchange format (SEF, Brunet et al., 2020), except that pressure data could not be corrected to 0°C as no temperature data are available. Temperature was digitised, but not converted, as the scale is unknown. The quality control (QC) package datarescqc (Brugnara et al., 2019) was used (see also descriptions in Brugnara et al., 2020; Brunet et al., 2020). When a quality issue was found, this was noted in the column "Meta" of the SEF file (the measurement value itself was not changed). As an example, Fig. 7 shows the digitised tabular data as well as the SEF file for the case of July 1721. The header of the SEF file provides metadata at the station level, including whether the pressure data were temperature corrected (PTC = N) or gravity corrected (PGC = Y) and the QC software used. Metadata on the level of individual measurements is indicated in the column meta. In this case, two values are flagged. The information "qc=wmo_time_consistency" means that the pressure change to the next measurement is larger than the threshold recommended by WMO (1993). In this case, the cause the low value at 11 pm local time (22:06 UTC), after which pressure jumps back to the previous value. Both jumps are flagged. The conversion from local solar time to UTC does not take into account the equation of time, therefore the difference with UTC is always 54 minutes.

| | Date | | | Time | | Thermometer | Barometer | | Wind | | in sunlight | |
|---|---|---|---|---|---|---|---|---|---|---|---|---|
| 1 | Template | Europe_T1_DE_Berlin_1721-1728_subdaily | | | | | | | | | | |
| 2 | Project | PALAEO-RA | | | | | | | | | | |
| 3 | Images | Europe_T0_DE_Berlin_1718-1728_subdaily.pdf | | | | | | | | | | |
| 4 | Pages | 334 – 349; 359 – 601 | | | | | | | | | | |
| 5 | **Before starting read the instructions in the other sheet!** | | | | | | | | | | | |
| 6 | | Date | | | Time | | Thermometer | Barometer | | Wind | | in sunlight | |
| 7 | Year | Month | Day | hour | v / n | | inches | lines | direction | force | | Notes of the digitiser |
| 8 | 1721 | 7 | 16 | 7 | v | 13 | 28 | 7.5 | | | | |
| 9 | | | | 12.25 | | 14 | 28 | 7.5 | | | | |
| 10 | | | | 5.75 | n | 14 | 28 | 7.5 | | | | |
| 11 | | | | 11 | n | 12.5 | 28 | 0.5 | | | | |
| 12 | | | 17 | 8 | v | 13 | 28 | 7.5 | | | | |
| 13 | | | | 1 | n | 14 | 28 | 7.25 | | | | |
| 14 | | | | 5.25 | n | 14.25 | 28 | 6.75 | WNW | | | |
| 15 | | | | 10 | n | 12.75 | 28 | 6.5 | | | | |

```
SEF       1.0.0
ID        Europe_Berlin_1
Name      Berlin
Lat       52.5656
Lon       13.3106
Alt       36
Source    PALAEO-RA
Link      NA
Vbl       p
Stat      point
Units     hPa
Meta      Observer=Kirch | PTC=N | PGC=Y|QC software=dataresqc v1.1.0
Year   Month   Day   Hour   Minute   Period   Value    Meta
1721   7       16    6      6        0        998.8    orig.time=7v|orig=28.7.5Rh.in
1721   7       16    11     21       0        998.8    orig.time=12.25|orig=28.7.5Rh.in
1721   7       16    16     51       0        998.8    orig.time=5.75n|orig=28.7.5Rh.in|qc=wmo_time_consistency
1721   7       16    22     6        0        978.4    orig.time=11n|orig=28.0.5Rh.in|qc=wmo_time_consistency
1721   7       17    7      6        0        998.8    orig.time=8v|orig=28.7.5Rh.in
1721   7       17    12     6        0        998      orig.time=1n|orig=28.7.25Rh.in
1721   7       17    16     21       0        996.6    orig.time=5.25n|orig=28.6.75Rh.in
1721   7       17    21     6        0        995.9    orig.time=10n|orig=28.6.5Rh.in
```

**Fig. 7.** Example of the digitised data in the spreadsheet and the corresponding data in the SEF format.

The data were submitted to the GLAMOD data base of Copernicus Climate Change Service (Noone et al., 2021). All data, both temperature and pressure, are also published as spreadsheets as a supplement to this article.

For assessing the Berlin series, we used pressure series from the HCLIM database (Lundstad et al., 2022). Specifically, we selected monthly data for pressure for all stations in Europe with at least 30 months of overlap (the shortest overlap is 45 months). In addition, daily pressure data from Nuremberg were used. Further, we used Gdansk temperature (in the form monthly minima and monthly maxima). Finally, we used sea-level pressure from the reconstruction EKF400v2 (Valler et al., 2022) at the grid point closest to Berlin for comparison with the digitised measurements from Berlin (note that Berlin pressure was not used for the reconstruction and hence is independent).

## 5. Results

*5.1. Digitising and processing*

We digitised 42065 pressure measurements and 39639 temperature measurements. Of the former, 49 have a flag "wmo_time_consistency" and two have a flag "duplicate_observation_time". An overview

of the temporal coverage of pressure and temperature data is given in Fig. 8. The data cover the periods 1730 to 1751 and 1756 to 1770 very well, with typically 3 to 4 measurements per day. The 1720s and the period 1752-5 are less well covered. No data are available for the years 1771-1773 and only few pressure measurements for 1774.

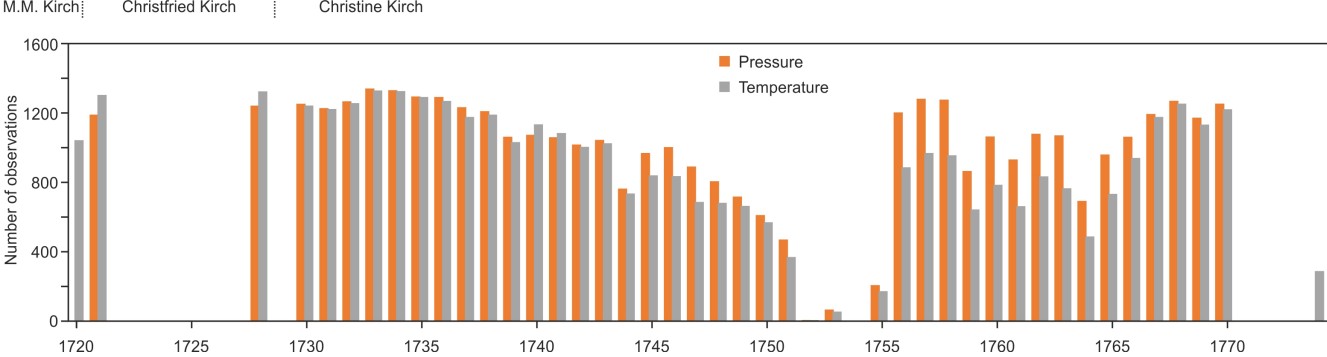

**Fig. 8.** Number of individual measurements digitised.

The time of day of the measurement is always indicated. A histogram of the measurement hours (Fig. 9, top) shows clear peaks, namely between 7 and 9 UTC, between 11 and 12 UTC, and 20 and 22 UTC. A smaller but distinct peak occurs between 14 and 15 UTC. Thus, the measurements were taken rather regularly at the usual observing times (local time is approximately 1 hour ahead of UTC).

A histogram of all pressure measurements is given in Fig 9 (bottom). An open question concerns the scale. Pelz (1978) assumes that pressure was not given in the official local scale, Rhineland inches (or Prussian inches, which is the same), as the numbers would otherwise be too low, but in Paris inches. However, assuming Paris inches would lead to numbers that are too high, the difference being 30 hPa. We assumed Rhineland inches, and consequently the average pressure is around 995 hPa, which is too low. We note that homogenization will be necessary to scale the pressure data. That said, the time series of monthly mean pressure (Fig. 10), does not show any obvious systematic change over time.

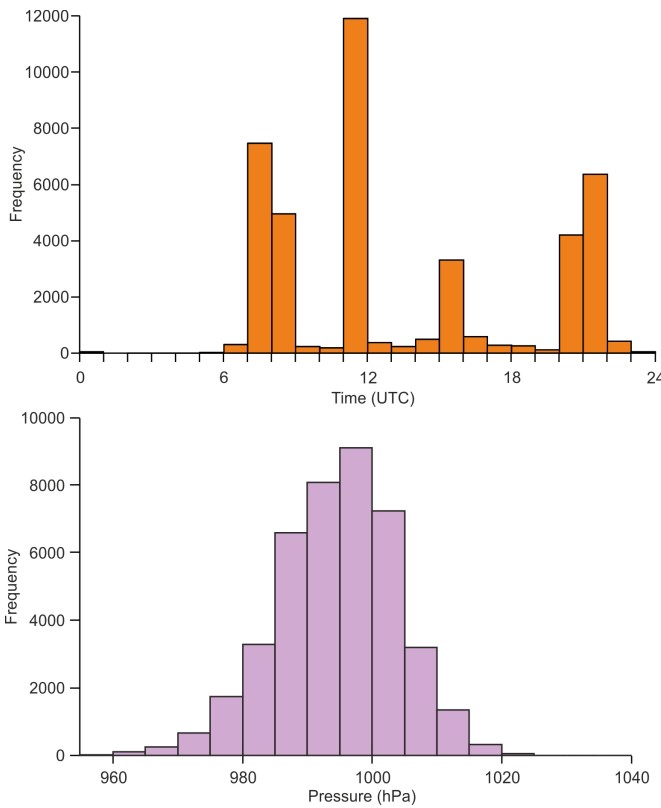

239

**Fig. 9.** Top: Histogram of the time of day (UTC) of measurements. Bottom: Histogram of pressure

measurements.

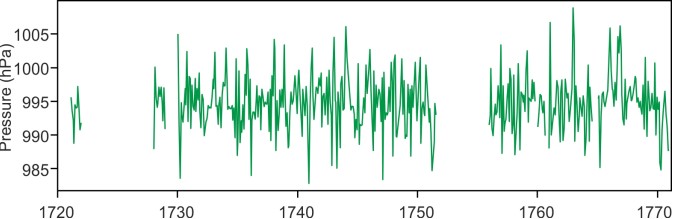

243

**Fig. 10.** Time series of monthly mean pressure.

*5.2. Comparison with other sources*

First we analysed the pressure data, which we compared with neighbouring series. Correlations on the

scale of monthly mean values are shown in Fig. 1 (right). Note that only few series (Paris and London)

have been homogenised. Most other series are analysed here in their original form. Nevertheless,

results show very high correlations exceeding 0.8 with Nuremberg and Zwanenburg. Also, the

correlations at Berlin with EKF400v2 reaches almost 0.8 and the correlation with London pressure is

in the same range. Somewhat lower, but still high correlations (>0.65) are found for Gdansk, Paris,

and Basel. Lower correlations exclusively originate from comparisons with shorter records which are

not homogenised and which have not yet been used much in the scientific literature, if at all. Overall

this clearly shows that the Berlin pressure data is of high relative quality and thus adds information to
the existing body of climate data.
*5.3. Case study: Particularly cold winter 1739/40*
One of the coldest European winters of the second millennium was 1739/40 (Schlaak, 1984,
Luterbacher and Wanner, 2002, Casty et al., 2005). It exhibits the lowest cold-season temperature of
the northern extratropical land areas of the last three centuries in a recent reconstruction by Reichen et
al. (2002). In fact, results from the spring phenology data from Europe used in that study are
summarized in Table 2. Clearly, the spring was extremely late in 1740, although it set the record only
in one of the series. In Figure 11 we present a very simple analysis of daily mean temperature in Berlin
from 1738 to 1743. For comparison we also show the monthly minimum and maximum temperatures
from Gdansk, which is over 400 km away. For Berlin, all measurements made on a particular day were
averaged without considering possible variations in the time of day of the measurement. Note also that
we do not know the scale of temperature, although it is reported that Christfried and Christine Kirch
used a Fahrenheit thermometer. The fact that slightly negative values are reached in Jan. 1740 might
indicate Fahrenheit temperature (where zero corresponds to -17.8 °C) or it might indicate that the
liquid dropped below a self-defined scale. The summer values are clearly too low to be degrees
Fahrenheit. Despite all these factors and despite the distance between the two sites, we clearly see
common variations. For instance, minimum temperatures were low in Nov. 1739, but then high in in
Dec. 1739 (with high maximum temperatures as well), then Jan. 1740 had very low minimum
temperatures but rather normal maximum temperature.

**Table 2.** Phenological spring data in Europe (from Reichen et al., 2022), rank of year 1740 and number of years
in the record.

| Location | Proxy | rank | N |
|---|---|---|---|
| Mälaren | ice break-up | 19 | 288 |
| Tallinn | ice break-up | 21 | 363 |
| Tornio | ice break-up | 4 | 311 |
| St. Peterburg | ice break-up | 6 | 173 |
| Zurich | cherry blossom | 1 | 283 |
| Haarlem | days of freeze | 5 | 147 |
| Turku | ice break-up | 2 | 79 |


For pressure, we can compare the Berlin series with that from Nuremberg (the observer was Johann
Gabriel Doppelmayr, who is also the author of the Atlas from which Fig. 3 is taken). The agreement is
very good, with a correlation coefficient of 0.83 despite some probable outliers. For the winter
1739/40, we find low pressure in Dec. 1739, but then high pressure in Jan. 1740. This is consistent
with temperatures.

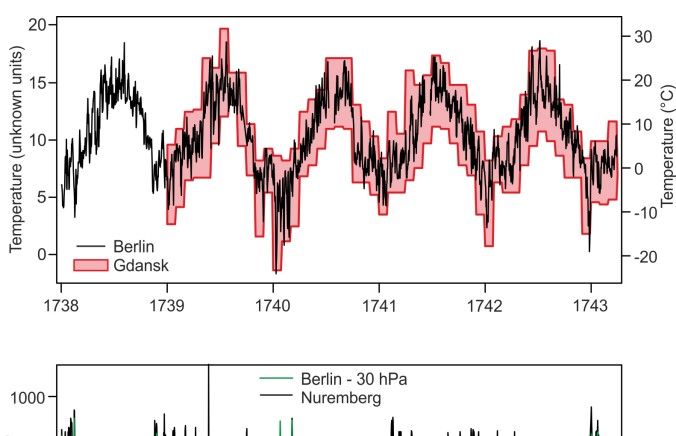

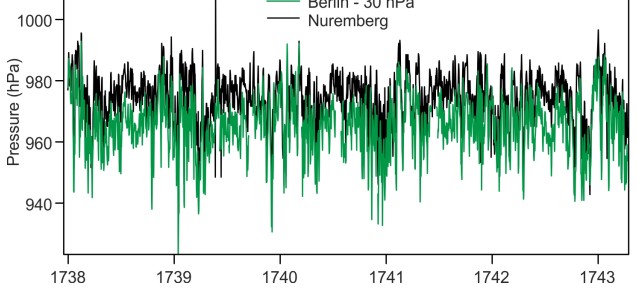


**Fig. 11.** Top: Daily averages of temperature from 1738-1742 by Christine Kirch (left axis; note that the temperature scale is not known). Also shown are the monthly minimum and maximum temperatures in Gdansk (right axis). Bottom: Daily mean pressure in Berlin (reduced by 30 hPa for comparison) and Nuremberg.

## 6. Discussion and conclusions

The Kirch family (Gottfried Kirch, his wife Maria Margaretha Kirch (née Winkelmann), their son Christfried Kirch and their daughter Christine Kirch) noted the weather for almost a century, from 1677 to 1774. From 1691 onward they were using a thermometer, later also a barometer, although regular observations only start in 1701. This body of measurements makes up the first part of the longest meteorological record in Germany. We imaged the diaries, totalling 10445 images, and make them available on a repository. Moreover, we digitised ca. 82000 instrumental observations after 1720 and publish them. Pressure data could be processed (although a reduction to 0 °C could not be performed). Temperature data could not be reduced because of an unknown (and changing) scale but were nevertheless digitised and are published as a supplement to this paper. Comparisons with other data suggest that the pressure series is trustworthy on the daily as well as monthly scale, although the scale remains uncertain. In fact, correlations with neighbouring series are very high. A brief analysis of the cold winter of 1739/40 suggests that also temperature measurements may contain useful information, even though the scale remains unknown. The newly digitised series will serve as the starting point for a new, daily Berlin series of temperature and pressure. The Kirch data set will be concatenated with other Berlin series from the 18th century currently under digitisation (Lambert, Jablonski, Gronau, Brand, and others) and homogenised to generate a more complete Berlin series.

The early Berlin data fall into a period in which not many other records are known and therefore they provide valuable information. However, there are some records with which the Berlin record can be compared. This includes long daily time series from Paris (Pliemon et al., 2022, Cornes et al., 2012a), London (Cornes et al., 2012b), Zwanenburg/DeBilt, shorter series from Nuremberg, Basel, Geneva, Zurich and Bern (e.g., Brugnara et al., 2022), St. Peterburg, Uppsala, and several Italian series from the IMPROVE project (Camuffo and Jones, 2002), among others (see inventory by Brönnimann et al., 2019). However, also other, non-instrumental weather diaries may be a good resource for comparisons, including those from Nuremberg (Brönnimann, 2023), Wroclaw (Przybylak, and Pospieszyńska, 2010), Gdansk (Filipiak et al., 2019) or Zurich (see weather diaries in EURO-CLIMHIST, Pfister et al., 2017). All sources taken together may provide a detailed view of weather in the 18th century.

**Funding.** The work was funded by the Swiss National Science Foundation project WeaR (188701) and the European Commission through H2020 (ERC Grant PALAEO-RA 787574).

**Acknowledgements.** We thank the FU Berlin for allowing us to photograph their material, the Paris Observatory, and the Crawford Collection of the Royal Observatory Edinburgh for allowing us to publish the data under a CC-BY-NC licence. We also thank Klaus-Dieter Herbst for his support and information on the Kirch family. In particular, we thank the students who performed the digitisation work.

**Data availability**

The images of the Kirch diaries can be found at: https://doi.org/10.48620/222

SEF data files for pressure in Berlin as well as Nuremberg have been submitted to the GLAMOD data base of Copernicus Climate Change service, they are also attached to this submission, together with the raw files for all variables for Berlin.

The monthly pressure data are part of the H-CLIM collection: https://doi.pangaea.de/10.1594/PANGAEA.940724

**Author contributions**

SB imaged the diaries, YB organised the digitisation, performed the quality control and all processing and formatting steps of the Berlin data. SB performed the analyses in the paper. Both authors wrote the paper.

**Competing interests.** The contact author has declared that none of the authors has any competing interests.

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
