# Peer review of "The weather diaries of the Kirch family: Leipzig, Guben, and Berlin, 1677- # 2 1774"

_Climate of the Past, 2023_

## Author Response (AR1)

**Reply to Reviewer 1:**

The author collected the weather diaries of the Gottfried Kirch family, photographed them all, digitized the data since 1720, briefly processed and analyzed the barometric data, and made a case for the extreme cold winter of 1739-1740. The work is of great significance to historical climatology research in Germany.

I believe that the authors likely have valuable high-resolution (daily-scale) weather observations data that are worthy of a publication. After reading the manuscript, I found that the author's description was relatively simple and there were many ambiguities. Therefore, I suggest that a **major revision** is needed before the official publication.

- In the Chapter 2 & 3, the authors describe in detail the life of the family members, as well as the time periods they each recorded and the tools they used. But unfortunately, I did not get more information about the content of the observations: for example, what were the conditions of the observations, what was the routine of the observations, how many times a day were the observations made, and were they timed? How to ensure the consistency of observation data after the shift of observation position. In-depth descriptions of these elements can increase the scientific validity of the data and allow later users to use the record more confidently.

  Thanks. The revised manuscript adds more details. Yes, we have the time of observation, and we will show a histogram of the times. There are three very strong peaks (morning, noon, evening) and a fourth, somewhat smaller peak in the mid-afternoon. Concerning the shift in observation sites: The data presented in this article start in 1721, so after the mentioned shift. They were presumably all taken at location B. We add a "family tree" figure on the observers (suggestion of reviewer 2) and to this also add the locations.

- In the method section of Chapter 4, the authors' description is rather brief, could they add a more detail description about the process of digitizing the data. How exactly was it extracted and what information was extracted? How does data quality control work: were there any outliers or missing measurements or ambiguous values recorded, and how were they handled if there were? In addition, I suggest that the authors give an example of the digitized results, such as a tabular presentation, which would provide more insight into what the authors did with the data.

  Thanks, we will make this much more clear.

  In terms of digitising, the supplementary material does not only contain all digitised data but it also contains the instructions given to the digitizers (as a separate Tab called "Info" in the spreadsheet). A template was handed to the digitizers (header of Tab "Template"), which was then filled. The digitizers also had the possibility to write comments in the spreadsheet (e.g., when a value was unreadable).

  Temperature data were not further processed. For the pressure data we used the R package dataresqc (Brugnara et al., 2019, Brunet et al., 2020) that was developed in the Copernicus Climate Change Service.

Brugnara, Y., Gilabert, A., Ventura, C., and Hunziker, S.: dataresqc: Quality control tools for climate data developed by the C3S Data Rescue Service, available at: https://github.com/c3s-data-rescue-service/dataresqc, last access: 10 September 2019.

The results of the qc are indicated in the formatted SEF file in the column meta. This indicates which qc test within dataresqc was not fulfilled. No value is excluded.

In the revised manuscript we show an additional figure with the tabulated data (spreadsheet) and a figure with an excerpt of the SEF formatted data that includes a case with a QC flag.

Furthermore, we add the paragraph:

We digitised temperature and pressure measurements. The actual keying work was performed by students of the University of Bern, who received a template and instructions (see spreadsheets in Supplementary Data: Tab "Info" contains the instructions).

The pressure data were then processed as described in Brugnara et al. (2020) and converted to the station exchange format (SEF, Brunet et al., 2020), except that pressure data could not be corrected to 0°C as no temperature data are available. Temperature was digitised, but not converted, as the scale is unknown. The quality control (QC) package datarescqc (Brugnara et al., 2019) was used (see also descriptions in Brugnara et al., 2020; Brunet et al., 2020). When a quality issue was found, this was noted in the column "Meta" of the SEF file (the measurement value itself was not changed). As an example, Fig. 7 shows the digitised tabular data as well as the SEF file for the case of July 1721. The header of the SEF file provides metadata at the station level, including whether the pressure data were temperature corrected (PTC = N) or gravity corrected (PGC = Y) and the QC software used. Metadata on the level of individual measurements is indicated in the column meta.  In this case, two values are flagged. The information "qc=wmo_time_consistency" means that the pressure change to the next measurement is larger than the threshold recommended by WMO (1993). In this case, the cause the low value at 11 pm local time (22:06 UTC), after which pressure jumps back to the previous value. Both jumps are flagged. The conversion from local solar time to UTC does not take into account the equation of time, therefore the difference with UTC is always 54 minutes.

And later:

The time of day of the measurement is always indicated. A histogram of the measurement hours (Fig. 9, top) shows clear peaks, namely between 7 and 9 UTC, between 11 and 12 UTC, and 20 and 22 UTC. A smaller but distinct peak occurs between 14 and 15 UTC. Thus, the measurements were taken rather regularly at the usual observing times (local time is approximately 1 hour ahead of UTC).

- For the results section in Chapter 5, I suggest that the authors have a more detailed description of the data, such as adding a simple statistical description of the observed

data (mean, maximum, etc.) in Chapter 5.1; the temporal variation characteristics of the reconstructed series can be analyzed, etc. In addition, the authors can perform some analysis of the trend of temperature and pressure changes within that century based on the available data, etc. In-depth description and use of the data allows the reader to have a more direct feeling of the data and to better promote the dataset.

Thanks, we add some overview statistics as well as a histogram of all individual pressure data and a time series plot of the monthly mean pressure data.

A paragraph is added

A histogram of all pressure measurements is given in Fig 9 (bottom). An open question concerns the scale. Pelz (1978) assumes that pressure was not given in the official local scale, Rhineland inches (or Prussian inches, which is the same), as the numbers would otherwise be too low, but in Paris inches. However, assuming Paris inches would lead to numbers that are too high, the difference being 30 hPa. We assumed Rhineland inches, and consequently the average pressure is around 995 hPa, which is too low. We note that homogenization will be necessary to scale the pressure data. That said, the time series of monthly mean pressure (Fig. 10), does not show any obvious systematic change over time.

- At the end of the manuscript, I hope the author can add some discussion to further analyze the value of this data set and look at other possible use of this data set in the future.

  Thanks, we add the comment that this data set will be concatenated with other Berlin series from the 18th century currently under digitisation (Lambert, Jablonski, Gronau, Brand, and others) to generate a more complete Berlin series .

I hope our comments will help the authors, whose work I believe to be of great scholarly value, and look forward to seeing a new manuscript from the authors.

**Reply to reviewer 2**

The weather diaries during historical times are important to reconstruct the climate parameters when the instrumental measurement was not started. The paper introduced the history of the weather diaries of the Kirch family and addressed the detailed recording process, unquestionably, this kind of diaries are valuable. If the temperature scale can be converted and keep the homogeneous of the series, that would be great. I just have several minor comments:

In section 2: Is it possible to plot a simple family tree, with year of measurement, and the readers could easily to get the information the year, and who measured? I think it is clearer than reading several paragraphs.

Excellent idea! We will add a figure with family tree, names and measurement periods, there we also indicate the locations (comment by rev. 1).

- In Fig.1 What did the circles mean? if they indicate the area, please mark with unit.

  In the caption we wrote: circle area indicates $n$, will be changed to: circle area indicates the number of measurements.$n$

- P9-10, the number of figures could be wrong in the text.

  The numbers are correct.